# LARGER LANGUAGE MODELS DO IN-CONTEXT LEARNING DIFFERENTLY

## ABSTRACT

We study how in-context learning (ICL) in language models is affected by semantic priors versus input–label mappings. We investigate two setups—ICL with flipped labels and ICL with semantically-unrelated labels—across various model families (GPT-3, InstructGPT, Codex, an internal model, and an instruction-tuned variant of the internal model). First, experiments on ICL with flipped labels show that overriding semantic priors is an emergent behavior of model scale. While small language models ignore flipped labels presented in-context and thus rely primarily on semantic priors from pretraining, large models override semantic priors when presented with in-context exemplars that contradict priors, despite the stronger semantic priors that larger models may hold. We next study *semantically-unrelated label ICL* (SUL-ICL), in which labels are semantically unrelated to their inputs (e.g., foo/bar instead of negative/positive), thereby forcing language models to learn the input–label mappings shown in in-context exemplars in order to perform the task. The ability to do SUL-ICL also emerges primarily with scale, and large-enough language models can even perform linear classification better than random guessing in a SUL-ICL setting. Finally, we evaluate instruction-tuned models and find that instruction tuning strengthens both the use of semantic priors and the capacity to learn input–label mappings, but more of the former.

## 1 INTRODUCTION

Language models can perform a range of downstream NLP tasks via *in-context learning* (ICL), where models are given a few exemplars of input–label pairs as part of the prompt before performing the task on an unseen example [2; 28, *inter alia*]. To successfully perform ICL, models can (a) mostly use semantic prior knowledge to predict labels while following the format of in-context exemplars (e.g., seeing "positive sentiment" and "negative sentiment" as labels and performing sentiment analysis using prior knowledge) and/or (b) learn the input–label mappings from the presented exemplars (e.g., finding a pattern that positive reviews should be mapped to one label, and negative reviews should be mapped to a different label).

Prior work on which of these factors drives performance is mixed. For instance, although Min et al. [25] showed that presenting random ground truth mappings in-context does not substantially affect performance (suggesting that models primarily rely on semantic prior knowledge), other work has shown that transformers in simple settings (without language modeling pretraining) implement learning algorithms such as ridge regression and gradient descent [1; 40; 10].

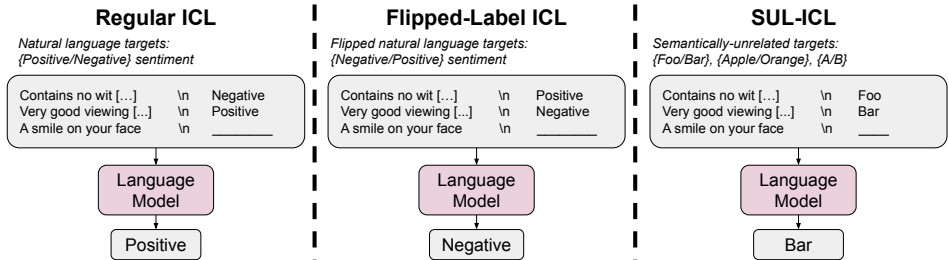

Figure 1: An overview of flipped-label ICL and semantically-unrelated label ICL (SUL-ICL), compared with regular ICL. Flipped-label ICL uses flipped targets, forcing the model override semantic priors in order to follow the in-context exemplars. SUL-ICL uses targets that are not semantically related to the task, which means that models must learn input–label mappings in order to perform the task because they can no longer rely on the semantics of natural language targets.

In this paper, we study how these two factors—semantic priors and input–label mappings—interact in several experimental settings (see Figure 1 for an example of each setting):

1. In **regular ICL**, both semantic priors and input–label mappings can allow the model to perform in-context learning successfully.
2. In **flipped-label ICL**, all labels in the exemplars are flipped, which means that semantic prior knowledge and input–label mappings disagree. Labels for the evaluation set stay the same, so for binary classification tasks, performing better than 50% accuracy in this setting means that the model is unable to override semantic priors, and performing below 50% accuracy means that the model is able to learn input–label mappings and override semantic priors.
3. In **semantically-unrelated label ICL** (SUL-ICL), the labels are semantically unrelated to the task (e.g., for sentiment analysis, we use "foo/bar" instead of "negative/positive"). Since the semantic priors from labels are removed, the model can only perform ICL by using input–label mappings.

We run experiments in these settings spanning multiple model families with varying sizes, training data, and instruction tuning (GPT-3, InstructGPT, Codex, an internal model, an instruction-tuned variant of the internal model) in order to analyze the interplay between semantic priors and input–label mappings,[1] paying special attention to how results change with respect to model scale. First, we examine flipped-label ICL, where we find that small models do not change their predictions when seeing flipped labels, but large models may flip their predictions to follow flipped exemplars (Section 3). This means that the behavior of overriding semantic priors with input–label mappings emerges with model scale, which should not be taken for granted because larger models presumably have stronger priors that are more challenging to override.

Second, we compare the SUL-ICL setting to regular ICL (Section 4). We find that small language models experience a large performance drop when semantic priors are removed, whereas large language models can perform the task well even without semantic priors from the labels. For some datasets, doing better than random in the SUL-ICL setting required substantial scaling (e.g., only the 540B internal model achieves above-random performance). We also found this to be true for high-dimensional linear classification tasks (Section 6). This means that learning input–label mappings without being given priors is also an emergent ability of large language models for those tasks.

Finally, we study the effect of instruction tuning [24; 46; 7] on ICL abilities (Section 5). We find that instruction-tuned models achieve better performance than pretraining-only models on SUL-ICL settings, which means that instruction tuning increases the model's ability to learn input–label mappings. On the other hand, we also see that instruction-tuned models are more reluctant to follow flipped labels, which means that instruction tuning decreases the model's ability to override semantic priors more than it increases its ability to learn input–label mappings. Overall, our work aims to shed light on the interaction between semantic prior knowledge and input–label mappings while considering the effects of scaling and instruction tuning.

---

[1]Many factors can affect ICL, including majority-label bias and recency bias [50]. We mitigated these biases by providing equal exemplars per class and randomizing the order of input–label pairs. We studied additional factors in Appendix C.3, Appendix C.4, Appendix C.5, and Appendix C.6. It is still possible, however, that other factors could be at play, though we believe that the major factors being analyzed are the two described.

# 2 EXPERIMENTAL SETUP

## 2.1 EVALUATION TASKS

We experiment on seven NLP tasks that have been widely used in the literature [16; 41; 42]. These evaluation tasks and an example prompt/target pair are shown in Figure 9 in the Appendix; additional dataset details are described in Appendix B. The seven tasks are: Sentiment Analysis [37, **SST-2**]; Subjective/Objective Sentence Classification [8, **SUBJ**]; Question Classification [20, **TREC**]; Duplicated-Question Recognition [5; 41, **QQP**]; Textual Entailment Recognition [9; 42, **RTE**]; Financial Sentiment Analysis [23, **FP**]; and Hate Speech Detection [26, **ETHOS**].[2]

## 2.2 MODELS

We perform experiments on five language model families as shown in Table 1. We use three families of OpenAI language models accessed via the OpenAI API: GPT-3 [2], InstructGPT [29], and Codex [4]. For GPT-3 models, ada, babbage, curie, and davinci seem to correspond to the following model sizes: 350M, 1.3B, 6.7B, and 175B [13]. For InstructGPT and Codex, however, it is not publicly

| Model Family | Model Name (Abbreviation) |
|---|---|
| GPT-3 | ada (a), babbage (b), curie (c), davinci (d) |
| InstructGPT | text-ada-001 (a-1), text-babbage-001 (b-1), text-curie-001 (c-1), text-davinci-001 (d-1), text-davinci-002 (d-2) |
| Codex | code-cushman-001 (c-c-1), code-davinci-001 (c-d-1), code-davinci-002 (c-d-2) |
| Internal language model | LLM-8B, LLM-62B, LLM-540B |
| Instruction-tuned internal language model | IT-LLM-8B, IT-LLM-62B, IT-LLM-540B |

Table 1: Models used in this paper.

known what the sizes of these language models are, but we assume that they are in increasing model scale for some scaling factor.

We also experiment on three different sizes of an internal language model (LLM-8B, LLM-62B, and LLM-540B) and their instruction-tuned variants (IT-LLM-8B, IT-LLM-62B, IT-LLM-540B). Our internal language models have the same training data and protocol and only differ by model size, which provides an additional data point for the effect of scaling model size specifically. Because many experiments rely on querying OpenAI models that are not publicly-available, we do not report the compute used for these experiments.[3]

## 2.3 ADDITIONAL EXPERIMENTAL DETAILS

As additional experimental details, we follow the prior literature on in-context learning and use a different set of few-shot exemplars for each inference example [2; 6; 44, *inter alia*]. By default, we use $k = 16$ in-context exemplars per class, though we also experiment with varying number of exemplars in Section 4 and Appendix D.2. We also use the "Input/Output" template for prompts shown in Figure 9, with ablations for input format shown in Appendix C.4 and Appendix C.5, and the semantically-unrelated "Foo"/"Bar" targets as shown in Figure 9 (ablations for target type are shown in Appendix C.3). Finally, to reduce inference costs, we use 100 randomly sampled evaluation examples per dataset, as it is more beneficial to experiment with a more-diverse range of datasets and model families than it is to include more evaluation examples per dataset, and our research questions depend more on general behaviors than on small performance deltas (note that all $y$-axes in our plots go from 0%–100% accuracy).

---

[2]In preliminary experiments (Appendix C.3), we also tried two additional tasks: Question–Answering [32; 41, **QNLI**] and Coreference Resolution [18; 42, **WSC**], but even the largest models had very weak performance on these tasks in many settings, so we do not include them in further experimentation.

[3]We used internal resources to evaluate our internal language models, so we do not report these numbers in order to retain anonymity.

## 3 INPUT–LABEL MAPPINGS OVERRIDE SEMANTIC PRIORS IN LARGE MODELS

To what extent do models override semantic priors from pretraining in favor of input–label mappings presented in-context? When presented in-context exemplars with flipped labels, models that override priors and learn input–label mappings in-context should experience a decrease in performance to below random guessing (assuming ground-truth evaluation labels are not flipped).

To test this, we randomly flip an increasing proportion of labels for in-context exemplars. As shown in Figure 1, for example, 100% flipped labels for the SST-2 dataset would mean that all exemplars labeled as "positive" will now be labeled as "negative," and all exemplars that were labeled as "negative" will now be labeled as "positive." Similarly, 50% flipped labels is equivalent to random labels, as we use binary classification datasets (we exclude TREC from this experiment since it has six classes). We do not change the labels of the evaluation examples, so a perfectly-accurate model that overrides priors should achieve 0% accuracy when presented with 100% flipped labels.

Figure 2 shows average model performance for each of the model families across all tasks with respect to the proportion of labels that are flipped (per-dataset results are shown in Figure 16). We see that there is a similar trend across all model families—at 0% flipped labels (i.e., no labels are changed), larger models have better performance than small models, which is expected since larger models should be more capable than smaller models. As more and more labels are flipped, however, the performance of small models remains relatively flat and often does not dip below random guessing, even when 100% of labels are flipped. Large models, on the other hand, experience performance drops to well-below random guessing (e.g,. text-davinci-002 performance drops from 90.3% with 0% flipped labels to just 22.5% with 100% flipped labels). Note that GPT-3 models remove semantic priors (i.e., perform at guessing accuracy) but does not override them (i.e., perform significantly worse than guessing), even when presented with 100% flipped labels. For this reason, we consider all GPT-3 models to be "small" models because they all behave similarly to each other this way.

These results indicate that large models override prior knowledge from pretraining with input–label mappings presented in-context. Small models, on the other hand, do not flip their predictions and thus do not override semantic priors (consistent with Min et al. [25]). Because this behavior of overriding prior knowledge with input–label mappings only appears in large models, we conclude that it is an emergent phenomena unlocked by model scaling [47].

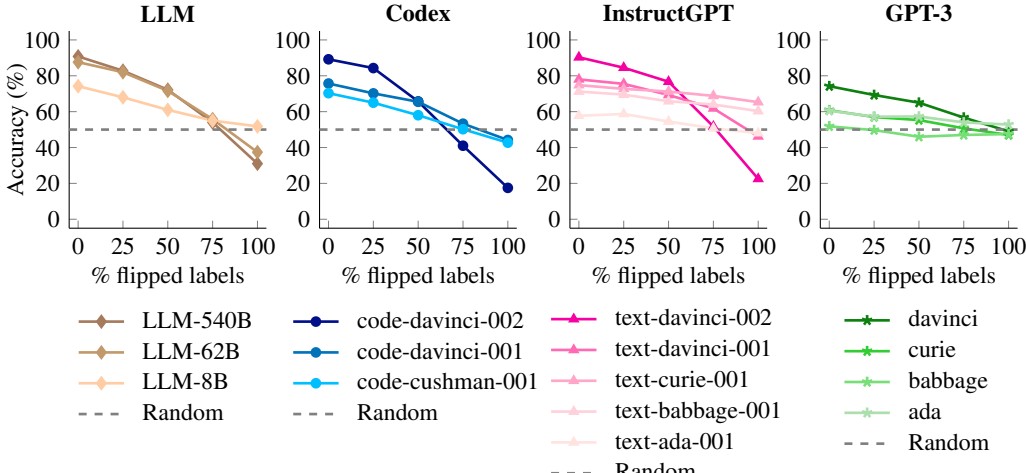

Figure 2: The behavior of overriding semantic priors when presented with flipped in-context exemplar labels emerges with model scale. Smaller models do not flip predictions to follow flipped labels (performance only decreases slightly), while larger models do (performance decreases to well below 50%). Ground truth labels for evaluation examples are not flipped, so if a model follows flipped labels, its accuracy should be below 50% when more than 50% of labels are flipped. For example, a model with 80% accuracy at 0% flipped labels will have 20% accuracy at 100% flipped labels if it flips its predictions. Accuracy is computed over 100 evaluation examples per dataset with $k = 16$ in-context exemplars per class and averaged across all datasets.

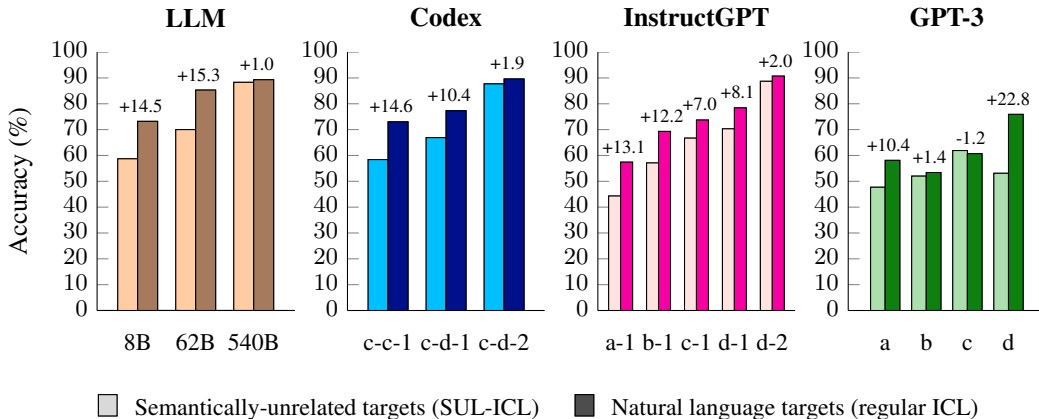

Figure 3: Small models rely more on semantic priors than large models do, as performance decreases more for small models than for large models when using semantically-unrelated targets instead of natural language targets. For each plot, models are shown in order of increasing model size (e.g., for GPT-3 models, a is smaller than b, which is smaller than c). We use $k = 16$ in-context exemplars per class, and accuracy is calculated over 100 evaluation examples per dataset and averaged across all datasets. A per-dataset version of this figure is shown in Figure 17 in the Appendix.

## 4 IN-CONTEXT LEARNING WITH SEMANTICALLY-UNRELATED LABELS CAN EMERGE WITH MODEL SCALE FOR SOME TASKS

Another way to examine how much models use semantic priors from pretraining versus input–label mappings is to replace natural language targets with semantically-unrelated targets. If a model mostly relies on semantic priors for in-context learning, then its performance should significantly decrease after this change, since it will no longer be able to use the semantic meanings of targets to make predictions. A model that learns input–label mappings in-context, on the other hand, would be able to learn these semantically-unrelated mappings and should not experience a major drop in performance.

We use an experimental setup that we call Semantically-Unrelated Label In-Context Learning (SUL-ICL) to test model behavior in these scenarios.[4] In this setup, all natural language targets are swapped with semantically-unrelated targets (we use "Foo" and "Bar" by default, although we get similar results with other semantically-unrelated targets—see Appendix C.3). For example, SUL-ICL relabels examples labeled as "negative" as "foo" and examples labeled as "positive" as "bar" for the SST-2 dataset (Figure 1). We then examine model performance in the SUL-ICL setup (in Appendix C, we investigate other aspects of the SUL-ICL setup such as remapping inputs, formatting prompts differently, changing target types, and using out-of-distribution datasets).

In Figure 3, we examine average model accuracy across all tasks on the SUL-ICL setup compared with a regular in-context learning setup (per-dataset results are shown in Figure 17). As expected, we see that increasing model scale improves performance for both regular in-context learning and SUL-ICL. The performance drop from regular ICL to SUL-ICL, however, is far more interesting. We find that using semantically-unrelated targets results in a greater performance drop from using natural language targets for small models compared with large models. Because small models are heavily affected when the semantic meaning of targets is removed, we conclude that they primarily rely on the semantic meaning of targets for in-context learning rather than learn the presented input–label mappings. Large models, on the other hand, experience very small performance drops after this change, indicating that they have the ability to learn input–label mappings in-context when the semantic nature of targets is removed.[5] Hence, the ability to learn input–label mappings in-context without being given semantic priors can also be seen as an emergent ability of model scale.

---

[4]Rong [34] previously evaluated a setup where they replaced natural language targets with non-alphanumeric characters; our paper uses a similar setup and investigates with more-extensive experimentation.

[5]For the reasons stated in Section 3, we consider davinci to be a small model.

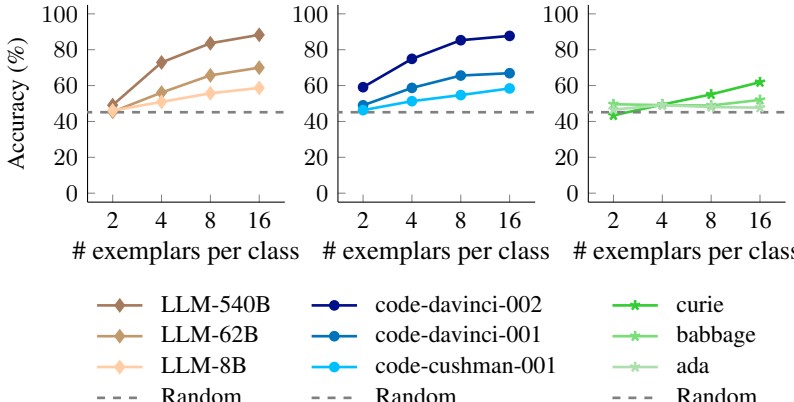

Figure 4: In the SUL-ICL setup, larger models benefit more from additional exemplars than smaller models do. Accuracy is calculated over 100 evaluation examples per dataset and averaged across all datasets. A per-dataset version of this figure is shown in Figure 18 in the Appendix.

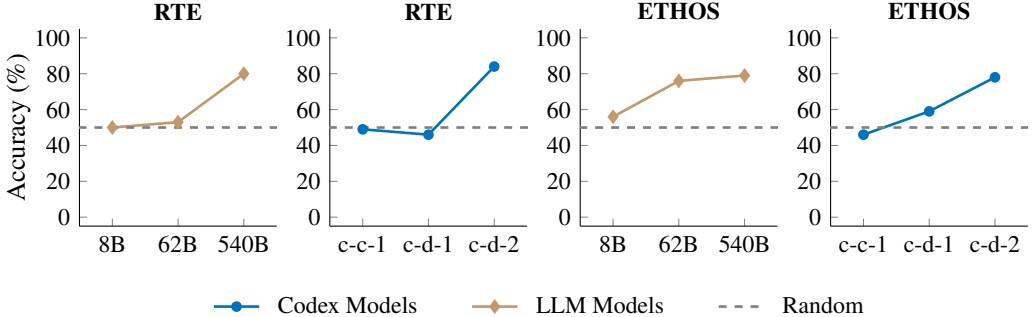

Figure 5: Some tasks in the SUL-ICL setting emerge with scale and can only be successfully performed by large-enough models. These experiments use $k = 8$ in-context exemplars per class. Accuracy is calculated over 100 evaluation examples.

We next analyze how models perform on a SUL-ICL setup when presented with an increasing number of in-context exemplars, and we show these data in Figure 4 (per-dataset results are shown in Figure 18). We find that for the three model families that we tested,[6] including more in-context exemplars results in a greater performance improvement for large models than it does for small models. This indicates that large models are better at learning from in-context exemplars than small models are, implying that large models are more capable of using the additional input–label mappings presented in context to better learn the correct relationships between inputs and labels.

Finally, looking at the per-dataset performance reveals how the ability to perform some benchmark tasks in the SUL-ICL setting emerges with scale. In Figure 5, we highlight two tasks (RTE and ETHOS) that seem particularly emergent in the SUL-ICL setting by plotting model performance at each model size for Codex and LLM models (Figure 18 shows how each model performs for each dataset). We see that performance on the RTE dataset is around random for LLM-8B and LLM-62B, yet increases to well above random for LLM-540B. Similarly, the performance on both the RTE and ETHOS datasets is around random for code-cushman-001 and code-davinci-001, then jumps to $80\%+$ for code-davinci-002. LLM models seem to emerge earlier on the ETHOS dataset, however, as the performance spikes when scaling from LLM-8B to LLM-62B. For many datasets that do not show emergence, even small models can outperform random guessing without many in-context exemplars (e.g., on SST-2, TREC, SUBJ, FP). These results show another example of how, for some tasks, the ability to learn input–label mappings in-context without being given semantic priors is only emergent in large-enough language models.

---

[6]We do not run on InstructGPT models or davinci due to the cost of running the large volume of experiments.

# 5   INSTRUCTION TUNING WITH EXEMPLARS IMPROVES INPUT–LABEL MAPPINGS LEARNING AND STRENGTHENS SEMANTIC PRIORS

A popular technique for improving the performance of pretrained language models is to finetune them on a collection of NLP tasks phrased as instructions, with few-shot exemplars as part of the finetuning inputs [24; 46; 7; 21]. Since instruction tuning uses natural language targets, however, an open question is whether it improves the ability to learn input–label mappings in-context or whether it strengthens the ability to recognize and apply semantic priors, as both would lead to an improvement in performance on standard ICL tasks.

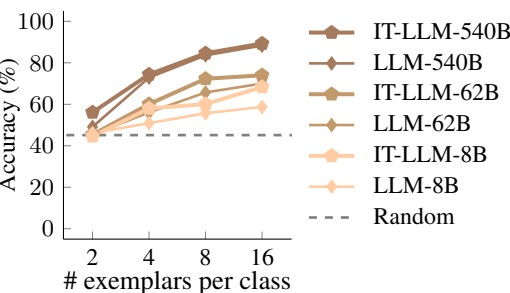

Figure 6: Instruction-tuned language models are better at learning input–label mappings in the SUL-ICL setting than pretraining-only language models are. Accuracy is calculated using 100 evaluation examples per dataset and averaged across six datasets. A per-dataset version of this figure is shown in Figure 19 in the Appendix.

To study this, we run the same experiments from Section 3 and Section 4, and we now compare LLM models to their instruction-tuned versions (IT-LLM). We do not compare InstructGPT against GPT-3 models in this experiment because we cannot determine if the only difference between these model families is instruction tuning (e.g., we do not even know if the base models are the same).

Figure 6 shows the average model performance across all datasets with respect to the number of in-context exemplars for LLM and IT-LLM models. We see that IT-LLM performs better in the SUL-ICL setting than LLM does, an effect that is most prominent in small models, as IT-LLM-8B outperforms LLM-8B by 9.6%, almost catching up to LLM-62B. This trend suggests that instruction tuning strengthens the ability to learn input–label mappings (an expected outcome).

In Figure 7, we show model performance with respect to the proportion of labels that are flipped for each LLM and IT-LLM model. We find that, compared to pretraining-only models, instruction-tuned models are worse at flipping their predictions—IT-LLM models were unable to override their semantics more than what could be achieved by random guessing, even with 100% flipped labels. Standard LLM models, on the other hand, could achieve as low as 31% accuracy when presented with 100% flipped labels. These results indicate that instruction tuning either increases the extent to which models rely on semantic priors when they are available or gives models more semantic priors, as instruction-tuned models are less capable of flipping their natural language targets to follow the flipped labels that were presented. Combined with the result from Figure 6, we conclude that although instruction tuning improves the ability to learn input–label mappings, it concurrently strengthens the usage of semantic priors, similar to the findings in Min et al. [24].

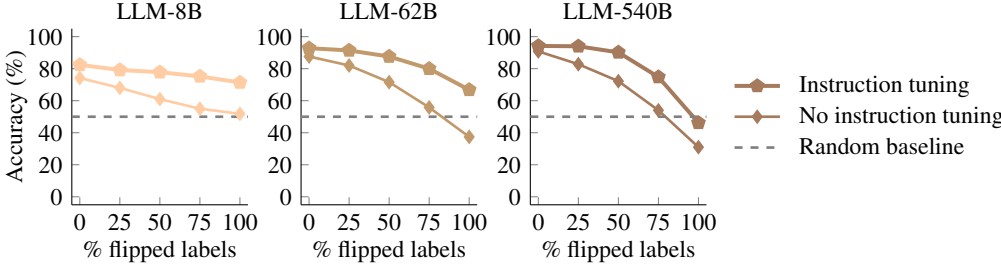

Figure 7: Instruction-tuned models are worse than pretraining-only models are at learning to override semantic priors when presented with flipped labels in-context. We use $k = 16$ in-context exemplars per class, and accuracy is calculated using 100 evaluation examples per dataset and averaged across six datasets. A per-dataset version of this figure is shown in Figure 20 in the Appendix.

## 6 LARGE LANGUAGE MODELS CAN PERFORM LINEAR CLASSIFICATION

In addition to the natural language reasoning abilities that we studied throughout the rest of the paper, we also seek to learn about how model scale affects the ability to perform other tasks. Specifically, we look at the linear classification task, where large models should perform better than small models (especially at high dimensions) if their greater capacity to learn input–label mappings as shown in Section 4 also holds for non-natural-language tasks.

To analyze this, we create $N$-dimensional linear classification datasets and examine model behavior with respect to the number of dimensions in the SUL-ICL setup. In these datasets, we provide $k$ $N$-dimensional points above a threshold and $k$ $N$-dimensional points below that same threshold as in-context exemplars, and the model must determine whether an $N$-dimensional evaluation point is above or below the threshold (we do not tell the model the equation or the threshold). When selecting random $N$-dimensional points, we use random integers between 1 and 1000 for each coordinate value. Algorithm 1 in the Appendix shows the precise dataset generation procedure.

In Figure 8, we show Codex model performance on $N = 16$ dimensional linear classification (per-dimension results on Codex and LLM models are shown in Figure 21 in the Appendix). The largest model outperforms random guessing by 19% on this task, while smaller models cannot outperform random guessing by more than 9%, suggesting that there exists some scaling factor that allows large-enough language models to perform high-dimensional linear classification.

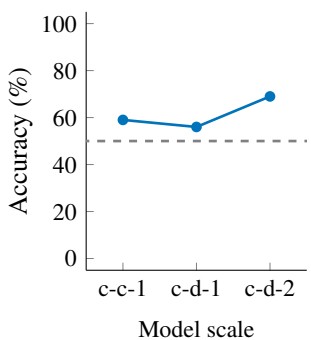

Figure 8: Successfully performing 16-dimensional linear classification emerges with model scale for Codex models. Accuracy is calculated over 100 evaluation examples with $k = 16$ in-context exemplars per class. Per-dimension results are shown in Figure 21 in the Appendix.

## 7 RELATED WORK

### 7.1 IN-CONTEXT DEMONSTRATIONS PROVIDE SEMANTIC PRIOR KNOWLEDGE

There has been a growing body of work on in-context learning that suggests that good performance is primarily driven by semantic priors and other factors such formatting and inducing intermediate token generation. For instance, Min et al. [25] showed the surprising result that using random ground-truth labels in exemplars barely hurts performance, suggesting that performance is instead mainly driven by the label space, distribution of input text, and overall format of the sequence. Along the same lines, Madaan & Yazdanbakhsh [22] and Wang et al. [43] show that for chain-of-thought prompting [48], logically-incorrect prompts do not hurt performance on multi-step reasoning tasks. On a theoretical level, Xie et al. [49] provide an explanation of in-context learning in which transformers infer tasks from exemplars because they are trained to infer latent concepts during pretraining, and prior knowledge obtained from pretraining data can then be applied to in-context examples. Finally, Reynolds & McDonell [33] showed that clever zero-shot prompts can outperform few-shot prompts, which implies that some NLP tasks benefit more from leveraging the model's existing knowledge than from learning about the task from in-context exemplars. In this paper, we do not contest the claim that language models can benefit greatly from semantic prior knowledge—our results instead add nuance to the understanding of ICL by showing that, when semantic prior knowledge is not available, large-enough language models can still do ICL using input–label mappings. Our experiments are consistent with Min et al. [25] for models scaling up to davinci, and we show that learning input–label mappings only emerges with larger models (e.g., LLM-540B, text-davinci-002, and code-davinci-002).

### 7.2 LEARNING INPUT–LABEL MAPPINGS

Other recent work has suggested to some degree that language models can actually learn input–label mappings from exemplars given in-context, which is a more-attractive ability than using semantic priors because it means that the model would be able to perform a wide range of tasks even if

those tasks are not seen in or even contradict pretraining data. For instance, transformers trained from scratch can perform in-context learning on linear-regression datasets with performance that is comparable to the least-squares estimator [14], and recent work has shown that transformers can do so by implementing standard learning algorithms such as ridge regression and gradient descent [1; 40; 10]. In the natural language setting, Webson & Pavlick [45] showed that language models learn just as fast with irrelevant or misleading prompts during finetuning or prompt-tuning. Our work makes similar claims about the ability for language models to learn tasks via input–label mappings only, though it differs crucially in that we observe frozen pretrained transformers without any additional learning.

### 7.3 EMERGENT PHENOMENA IN LARGE LANGUAGE MODELS

In this paper we have also focused on the effect of scaling on in-context learning, which relates to a nascent body of work showing that scaling language models leads to qualitatively-different behavior [12; 47; 38]. For instance, it has recently been shown that scaling up language models can allow them to perform a variety of challenging tasks that require reasoning [48; 6; 17; 51]. Our experimental findings on the flipped-label ICL setup show that language models can learn input–label mappings even when the input–label mapping contradicts the semantic meaning of the label, demonstrating another type of symbolic reasoning where language models can learn input–label mappings regardless of the actual identity of the labels. Although we have shown that this behavior is emergent with respect to model scale, the investigation of why scaling unlocks such behaviors [49; 3] is still an open question that we leave for future work.

## 8 LIMITATIONS

While our study sheds light on the interplay between semantic priors and input–label mappings in in-context learning for language models, there are several limitations to our work. An open question is how to apply our findings in a generative setting—we evaluated models on a range of classification tasks with discrete labels, but we did not test any generation tasks since it is unclear how to study the role of in-context demonstrations in those settings. Additionally, we examined the emergent ability of large language models to override semantic priors and learn input–label mappings. It is unknown, however, whether these emergent abilities may be affected by changes to the pretraining objective, architecture, or training process, and future work could investigate these factors. Moreover, as stated in Section 2.3, our experiments were conducted using only 100 evaluation examples per dataset because we prioritized using more datasets and model families over more evaluation examples per dataset. Future work could thus evaluate models on our settings using larger evaluation sizes per dataset. While we prioritized evaluating more model families, we note that our experiments in Section 5 were only conducted on LLM models, leaving open the question of whether the result generalizes to other model families as well.

## 9 CONCLUSIONS

In this paper, we examined the extent to which language models learn in-context by utilizing prior knowledge learned during pretraining versus input–label mappings presented in-context. We first showed that large language models may override semantic priors when presented with enough flipped labels (i.e., input–label mappings that contradict prior knowledge), and that this behavior emerges with model scale. We then created an experimental setup that we call Semantically-Unrelated Label In-Context Learning (SUL-ICL) which removes semantic meaning from labels by replacing natural language targets with semantically-unrelated targets. Successfully doing ICL in the SUL-ICL setup is another emergent ability of model scale. Additionally, we analyzed instruction-tuned language models and found that instruction tuning improves the capacity to learn input–label mappings but also strengthens semantic priors. Finally, we examined language model performance on linear classification tasks, finding that successfully performing high-dimensional linear classification emerges with model scale. These results underscore how the in-context learning behavior of language models can change depending on the scale of the language model, and that larger language models have an emergent ability to map inputs to many types of labels, a form of true symbolic reasoning in which input–label mappings can be learned for arbitrary symbols.

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
