# OpenReview forum: "Larger language models do in-context learning differently"
_ICLR.cc/2024/Conference — ICLR 2024 Conference Withdrawn Submission_

### Official Review · Reviewer_AA4b · 2023-10-30

**Soundness:** 1 poor
**Presentation:** 3 good
**Contribution:** 2 fair
**Rating:** 3
**Confidence:** 4

**Summary:**

This paper empirically investigates the behaviors of various LLMs while performing in-context learning (ICL). Its core contribution includes a series of experiments and conclusions. First, by comparing the models’ performance flipping the labels of the in-context learning examples, the paper finds that larger models are better at overriding semantic priors and concludes that such capabilities are emergent. Second, the paper investigates the models’ performance replacing the in-context demonstrations with unrelated ones. The results suggest that larger models do better in learning the mappings between input instances and unrelated labels, and the paper therefore concludes that the capability of learning from unrelated labels is also emergent. Finally, the paper finds that instruction finetuning strengthens both the models’ semantic priors and the capability of learning input-label mappings, but more for the former.

This paper stands out to me as an entirely empirical paper without any contribution to new methods or theories, which is different from typical ICLR papers I would anticipate reading. As such, I’m setting a higher bar while examining its scientific value, specifically in terms of clearly laying out the research questions, carefully designing the experiments to answer them, and drawing evidence-supported conclusions. The paper falls on these aspects, due to the reasons I will list below. Therefore, I’m leaning toward a rejection.

**Strengths:**

- The exploration of in-context learning is highly relevant and timely
- The findings in this paper are interesting and can be useful for future research
- Writing is very clear

**Weaknesses:**

- “Semantic prior” is vaguely defined. For a more precise quantification of "semantic prior," the authors might consider examining zero-shot learning performance instead of ICL with original labels.
- The capability to override prior knowledge is concluded “emergent”  (Fig 2); but if one takes a closer look at the scaling trend, it’s just larger models do better than smaller ones, without any phase changes.
- The conclusion on the instruction-tuned vs. pretrained models is flawed: without disclosing the instruction tuning data, it is impossible to attribute certain behaviors to instruction tuning as a model-building phase, instead of tuning on specific datasets.
- In order to study the impact of certain aspects of the models, e.g., size, instruction tuning, one needs to do control for confounders such as pretraining/finetuning data. However, the paper never attempts to do any controlled experiments. We all know that the mysterious organization that built the 540B model, which the authors are affiliated with, can afford to do controlled experiments with open-source models that are more transparent.
- I think it is wrong to conclude that “all GPT-3 models are small” because they underperform on ICL with flipped labels. The characterization of a model as "small" should be based on the number of parameters, rather than its performance on specific tasks, which can be influenced by a variety of factors.
- There are better tasks for studying LLMs

**Questions:**

None

---

### Official Review · Reviewer_K1hb · 2023-11-01

**Soundness:** 3 good
**Presentation:** 4 excellent
**Contribution:** 2 fair
**Rating:** 6
**Confidence:** 4

**Summary:**

This work explores how model scale affects an LLMs ability to adapt to different label surface forms, which challenge prior semantic task knowledge, during in-context learning on NLP tasks. The authors propose two label transformations:
- flipped-label ICL, in which the class label tokens for a binary task are swapped in the in-context exemplars
- semantically-unrelated label ICL, in which the class labels are replaces with toy tokens (i.e. "foo/bar")

To explore these label alterations, the authors study performance of 4 families of LLMs, over varying model scale; importantly, the authors leverage an internal LLM for which everything else but scale is controlled for, and whose capacity goes up to 540B parameters.
They study average performance across 7 binary NLP tasks.

On the flipped label setting, the authors study how model performance is affected as the percentage of flipped in-context labels increases. They find that smaller models react "worse" to label flips, in some cases still achieving over 50% accuracy even when 100% of the in-context labels are flipped, suggesting that the in-context label flips are ignored; larger models react much better, in some cases achieving below random performance when 100% of labels are flipped.

In the semantically-unrelated setting, the authors show that larger models experience a much smaller generalization gap between semantically-related labels and semantically-unrelated labels than small models do.

Finally, the authors explore how instruction-tuning affects these results, finding that instruction-tuned models (a) adapt to semantically unrelated labels better across all model sizes, but (b) react worse to flipped labels than standard LLMs.

**Strengths:**

- The paper raises an interesting question: how much does a model rely on semantic information learned during pre-training to perform in-context learning? This is an important question given the clear importance of in-context learning in ML, coupled with our current lack of understanding it's underlying mechanisms.
- The experiments presented to answer the question are simple, and very easy to follow and understand, as is their motivation. In fact, overall, the paper is extremely clear and well written, and it's core ideas and motivation are presented very clearly.
- The experiments consider models that are very large (up to 540B parameter model in terms of a model in a controlled setting)

**Weaknesses:**

While often simplicity in methods is a good thing, I have some concerns with how much we can take away from the flipped-label experiments due to the following concern:

Smaller models tend to have worse ICL performance in general. If a small model achieves 70% accuracy in the standard setting, then by switching it's labels we expect it to achieve at least 30% accuracy. Conversely, for a large model that achieves 90% accuracy, we expect it to achieve 10% accuracy if it flips it's labels. So, intuitively, we should actually expect the smaller models to have higher accuracy on flipped label settings, which indeed they do.

Clearly, this does not solely explain the results in section 3, because no models (not even the largest) flip 100% of their label predictions. However, it is certainly a confounding variable in our interpretation of figure 2. For instance, if LLM-8B achieves 80% on 0 flipped labels and 50% on 100% flipped labels, that could be because it flipped 30% of it's predictions, or because it flipped 70% of it's predictions (50% of the flips being on "correct" examples, and 20% being on "incorrect" examples). Conversely, LLM-540B could go from 90% accuracy to 30% accuracy by flipping 70% of it's predictions, all of which are correct. So both models actually flipped the same number of predictions.

I think this exact scenario is very unlikely, but nevertheless I do think this raises an issue with the way the results are presented and how it affects our take-aways. I think a simple metric to study that would fix this issue is the percentage of predictions that are flipped, rather than the overall model accuracy.

A somewhat related nitpick, I think the claim that larger models do in-context learning _differently_ may be a bit overstated, as the results don't seem clearly black and white to me.
It seems as though smaller models to adapt to the context labels to some extent, just less so than larger models; similarly, larger models don't completely adapt to the context labels, as there is often a slight gap (in both flipped and USL).
So it seems as though larger models just do in-context learning _better_, which is maybe not as surprising.

Finally, unrelated to the above:
I don't see mentioned the possibility that the instruction-tuned models are actually trained on the evaluation tasks, but it doesn't seem implausible to me. This may affect the final take-away from section 5 because we may be forcing the model to "unlearn" the task it was directly trained on, which seems distinctly different from overriding general semantic priors learned through training.

**Questions:**

How are model predictions being aggregated? Is all of the error due to the model predicting the incorrect class? Or can some of the error be explained by the model predicting a token that does not map to either class?

---

### Official Review · Reviewer_X31b · 2023-11-01

**Soundness:** 3 good
**Presentation:** 3 good
**Contribution:** 2 fair
**Rating:** 5
**Confidence:** 4

**Summary:**

The paper studies the in-context learning (ICL) capabilities of large language models investigating the impact of semantic priors and input-label mappings on ICL with models of different sizes.

In one set of experiments with binary classification tasks, the input-label mapping for support examples is flipped so that the model’s understanding of the semantics of the label is not the same as input-label mapping in the evaluation examples. The authors state that smaller models are unable to counteract the flipped labels, but larger models tend to follow the template of flipped support examples

In another setting, labels of tasks are substituted with labels that have no semantic connection, essentially random labels. They observe that small models fail at the task given no semantic priors while larger models are more capable of executing the tasks.

**Strengths:**

The experiments are well-structured and mostly comprehensive.

Experiments are carried out on a diverse range of model families and sizes. Authors investigated different in-context learning settings under GPT-3, InstructGPT, Codex, their internal model and their internal instruction tuned model.

The inclusion of varying numbers of in-context exemplars per class and instruction tuned models is insightful.

**Weaknesses:**

The experiments lack varying templates for in-context learning. There is a big degree of difference in ICL performance of models given different ICL templates. These varying templates would make the claims stronger.

The paper could benefit from providing more explanations for the observations made. For instance, it would be valuable to have a clearer understanding of the factors that contribute to the improvement in input-label mapping and semantic prior learning through instruction tuning LLMs.

The accuracy of tasks is reported using only 100 evaluation examples which could be considered as potentially insufficient for a comprehensive assessment. The authors also mention this in limitations.

**Questions:**

Regarding flipped-label ICL, It could be valuable to consider altering or incorporating a metric that measures the difference between correct-label ICL and flipped-label ICL performance. It is possible that the model’s performance on the task and its associated labels might not be strong from the outset. Do you have any insights or ideas on this matter?

What insights can be drawn from the findings on flipped and semantically unrelated ICL? In real-world scenarios and datasets, inputs and labels are typically somewhat semantically related. I believe providing and example that mirrors a more real-world scenario would be beneficial in motivating the study.

Could the prior exposure to the training and potentially test examples of the tasks, as encountered by the model during pretraining, influence the results and emergence?

---

### Official Review · Reviewer_fEqW · 2023-11-02

**Soundness:** 3 good
**Presentation:** 3 good
**Contribution:** 2 fair
**Rating:** 5
**Confidence:** 4

**Summary:**

The authors investigate how ICL abilities in LLMs change with model scale. They specifically examine the interplay between two factors that can drive ICL performance: 1) leveraging semantic priors from pretraining, and 2) learning input-label mappings from the examples provided in context.

**Strengths:**

1. To test the two factors that can drive ICL performance, the authors design two different experimental setups, including Flipped-label ICL
and SUL-ICL.  They perform extensive experiments across model families like GPT-3, Codex, and internal models of varying sizes and achieve several key findings.
2. The experiments are well-designed and highlight insightful differences between small and large language models. The results on the model scale are important - larger models can exhibit different reasoning abilities despite likely having even stronger semantic priors.
3. The paper is well-written and easy to follow. The authors present their experimental results effectively with simple but clear figures.

**Weaknesses:**

1. More analysis is needed to provide insight into why the behaviors emerge with scale. The authors observe intriguing phenomena but do not dig into potential explanations and solutions.
2. Limited investigation of generative tasks, most focus is on classification. Extending ideas to generative settings could be valuable.
3. Besides the well-designed label settings in ICL, it is also valuable to study other factors that could also impact the performance of ICL, such as adversarial attacks or perturbations into the demos.

**Questions:**

1. Have you ever tested the performance after supervised fine-tuning (SFT) of the models on the downstream tasks?